# Regional Representativeness Analysis of Ground-Monitoring PM$_{2.5}$ Concentration Based on Satellite Remote Sensing Imagery and Machine Learning Techniques

**Rui Luo, Meng Zhang *** and **Guodong Ma**

School of Human Settlements and Civil Engineering, Xi'an Jiaotong University, Xi'an 710049, China; viola30@stu.xjtu.edu.cn (R.L.); guodongma@stu.xjtu.edu.cn (G.M.)
* Correspondence: zhangmeng01@mail.xjtu.edu.cn; Tel.: +86-186-2903-4600

**Abstract:** The issue of urban air quality in China has become increasingly significant due to industrialization and rapid urbanization. Although PM$_{2.5}$ is the major air pollutant in most cities of northern China and has a direct negative impact on human health, there is a problem of under-representativeness at Chinese monitoring stations. In some cities, due to the relatively fewer national control stations and the fact that the stations are located closer to pollution sources, under the current assessment system, the monitoring data are not sufficient for the fairness of air quality assessment in different cities. In this article, the multispectral data of Landsat 8 data, air quality data, and meteorological data from ground monitoring stations have been integrated together and imported to different PM$_{2.5}$-estimation models established based on the multi-layer back propagation neural network (MLBPN), support vector regression (SVR), and random forest (RF), respectively. According to the evaluation indices of R$^2$, RMSE, and ME, the estimation model based on the MLBPN revealed the best PM$_{2.5}$ estimation results and was therefore employed for the regional representativeness analysis in the study area of Xi'an, Shaanxi, China. The annual average PM$_{2.5}$ concentration in the study area is depicted after error correction using Kriging interpolation, which can be further used to evaluate and analyze the representativeness of monitoring stations in Xi'an. By calculating the difference between the actual station annual average and the annual average of estimated PM$_{2.5}$ concentration in the whole region, it can be found that the regional annual average value of PM$_{2.5}$ in Xi'an is overestimated. To sum up, this article proposes a feasible method for the spatial positioning of the air quality monitoring stations to be established.

**Keywords:** machine learning; PM$_{2.5}$ concentration estimation; regional representativeness

## 1. Introduction

PM$_{2.5}$ is harmful to human health because it can easily enter the human body. Although the national ground monitoring stations can meet several requirements for environmental monitoring, there are still drawbacks such as inadequate representativeness, imperfect monitoring grids, limited monitoring coverage, and inconsistent monitoring standards. Therefore, it is of utmost importance to analyze the representativeness of the current monitoring stations.

Ground monitoring and remote sensing monitoring are two main categories of PM$_{2.5}$ monitoring techniques. Ground monitoring relies on stations to maintain round-the-clock continuous observation, allowing for the direct acquisition of PM$_{2.5}$ concentrations at various times. However, because they are dispersed and few in number, ground monitoring stations are unable to reliably collect data on the ongoing changes in the spatial and temporal distribution of PM$_{2.5}$. Wide data coverage, ease of access, low acquisition costs, and thorough coverage of spatial areas are all characteristics of satellite remote sensing imagery. Through satellite remote sensing monitoring, which enables the investigation and analysis of pertinent environmental concerns on a broader spatial scale, it is possible

to acquire the continuous variation of the spatial and temporal distribution of $PM_{2.5}$ concentration values [1,2]. As a result, a prominent topic in atmospheric remote sensing is the estimation of $PM_{2.5}$ concentrations from satellite remote sensing images [3].

Since the 1990s, satellite remote sensing techniques have been utilized to estimate $PM_{2.5}$ [4–9]. Wang and Christopher (2003) performed an inversion of $PM_{2.5}$ concentrations in Alabama based on MODIS AOD products and achieved a correlation coefficient of 0.70 between simulated and actual values [10]. In studies conducted by Engel-Cox et al. (2005) and Gupta et al. (2006) [11,12], high correlations were obtained between AOD and $PM_{2.5}$ concentrations, which laid the foundation for the inversion of $PM_{2.5}$ concentrations by AOD. Since then, the estimation of $PM_{2.5}$ concentrations from AOD has utilized more complex nonlinear regression models, such as the generalized additive model (GAM) and geographically weighted regression (GWR) [13–21].

In earlier investigations, AOD served as the primary foundation for retrieving $PM_{2.5}$ concentrations. Despite the fact that the AOD-based $PM_{2.5}$ estimation algorithm is widely used, the results of the $PM_{2.5}$ concentration estimation obtained by the AOD method are greatly influenced by the parameter settings and values, which are not only time-consuming to calculate but also significantly vary in different regions and are not applicable globally. Therefore, new methodological models for estimation investigations of $PM_{2.5}$ using remote sensing images still need to be investigated.

In recent years, some scholars have started to try to use machine learning methods to explore the correlation between $PM_{2.5}$ concentrations and the relevant bands of multi-spectral satellite remote sensing data and use them as the basis for inversion studies of $PM_{2.5}$ concentrations. For example, Zhang et al. (2019) constructed a $PM_x$ estimation model based on the back-propagation neural network commonly used in machine learning with the reflectance of different bands in Landsat 8 remote sensing images as the main basis and selected the Beijing area for testing, and they obtained more satisfactory estimation results [22]; Ma et al. (2022) tried to use multiple linear regression, K-nearest neighbor, support vector regression, decision tree, random forest, back-propagation neural network, and other different machine learning methods for $PM_{2.5}$ estimation of Landsat 8 remote sensing image data, and in the test with Hangzhou as the target area, the random forest method achieved the best estimation results [23].

However, although some progress has been made in $PM_{2.5}$ estimation based on machine learning, the current research mainly focuses on model building, and there are fewer studies on error correction and representative analysis of monitoring stations. Based on Google Earth Engine (GEE), this article obtained Landsat 8 satellite image data of Xi'an for five consecutive years from 2017 to 2021, combined with $PM_{2.5}$ concentration monitoring data from the China National Environmental Monitoring Center (CNEMC) and meteorological data issued by the China Meteorological Administration (CMA) and data released through the National Oceanic and Atmospheric Administration (NOAA) of the U.S. federal government. The three methods of support vector regression (SVR), the multi-layer back propagation neural network (MLBPN), and random forest (RF) have been employed for $PM_{2.5}$ estimation. By comparing the evaluation indices of $R^2$, RMSE, and ME to select the optimal model, the air pollutant concentrations in Xi'an were estimated, and after error correction using Kriging interpolation, they can be further used to evaluate and analyze the representativeness of monitoring stations. On this basis, a quantitative evaluation of regional representativeness is supplied by comparing the average value of the estimated pollutant concentration in the region with the average value of the pollutant concentration at the station, and a suggestion for station selection is provided.

## 2. Materials and Methods

### 2.1. Study Area

Xi'an is located in the central part of the Guanzhong region, between longitude 107.40°E to 109.49°E and latitude 33.42°N to 34.45°N, with 11 municipal districts and 2 counties. Xi'an is located in the southern part of the center of the Guanzhong Plain in

Shaanxi Province, which has an alluvial plain in the north and denuded mountains at higher elevations in the south. The general topography is high in the southeast and low in the northwest and southwest. The Guanzhong Plain is a basin with a static atmosphere in winter, which is not conducive to the diffusion of pollutants, but it is easy to form a pile-up effect, so the haze is more difficult to dissipate. The Qinling Mountains straddle the southern part of Xi'an, with ridges ranging from 2000 to 2800 m above sea level, and they are an important geographical dividing point between the north and south of China. As shown in Figure 1, there are 13 CNEMC stations and 1 CMA-NOAA station.

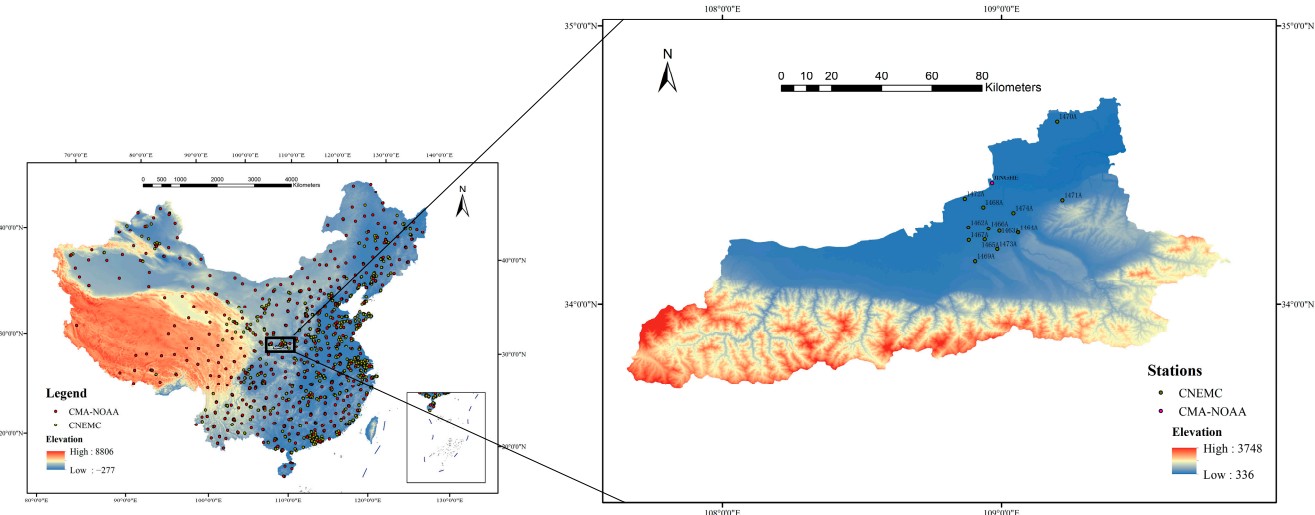

**Figure 1.** Distribution of CMA-NOAA and CNEMC stations in Mainland China and the study area.

### 2.2. Data Collection and Preprocessing

As shown in Table 1, the data used in this article mainly include the following:

- Satellite remote sensing data of Landsat 8;
- $PM_{2.5}$ concentration data released by the China National Environmental Monitoring Center (CNEMC);
- Meteorological data issued by the China Meteorological Administration (CMA) and released through the National Oceanic and Atmospheric Administration (NOAA) of the U.S. federal government.

**Table 1.** Data used in this article.

| Category | Variable | Unit | Resolution | Source |
|---|---|---|---|---|
| Landsat 8 | Landsat 8 OLI NDVI | Band Value | 16 day | GEE |
| $PM_{2.5}$ concentration data | $PM_{2.5}$ | $\mu g \cdot m^{-3}$ | 1 h | CNEMC |
| Meteorological data | WD WS PRS RH TEM | m/s ° Pa % °C | 3 h | CMA-NOAA |

### 2.2.1. Satellite Remote Sensing Data

The satellite data used in this project are Landsat 8 data. Landsat 8 is the eighth satellite in the U.S. Landsat program and was launched by NASA on 11 February 2013. Landsat 8 has two sensors, including TIRS and OLI. Landsat 8 has a spatial resolution of 30 m and a revisit period of 16 days in mainland China. Since its launch, Landsat 8 satellite data have been well used in the fields of land use type extraction, water environment monitoring, vegetation cover, biomass estimation, and surface temperature estimation. Based on some

of the literature published so far [22,24], B1 (0.43–0.45 μm), B3 (0.53–0.59 μm), B7 (2.11–2.29 μm), and the calculated NDVI based on the B4 (0.64–0.67 μm) and B5 (0.85–0.88 μm) of Landsat 8 are strongly relevant to the $PM_{2.5}$ concentrations. Therefore, B1, B3, B7, and the NDVI of Landsat 8 were selected as the primary input parameters for the $PM_{2.5}$ concentration estimations.

### 2.2.2. $PM_{2.5}$ Concentration Data

As a public institution directly under the Ministry of Ecology and Environment, PRC, the China National Environmental Monitoring Station (CNEMC) received approval to be built at the end of 1979 and was formally established in 1980. Its primary responsibilities include carrying out national ecological and environmental monitoring tasks, driving the development of technologies related to ecological and environmental monitoring, providing reports, monitoring data, and technical support for national environmental management and decision-making, and providing technical guidance for national ecological environment monitoring work.

As the technical center, network center, data center, quality control center, and training center of national ecological environment monitoring, the CNEMC provides ground monitoring data of urban air quality $PM_{2.5}$, $PM_{10}$, $SO_2$, $NO_2$, CO, and $O_3$, and publishes them to the public. There are more than 1500 CNEMC stations in Mainland China and 13 monitoring stations in Xi'an (Figure 1).

### 2.2.3. Meteorological Data

The National Oceanic and Atmospheric Administration (NOAA) is part of the U.S. Department of Commerce, which has five operational scientific units: the National Weather Service, the National Ocean Service, the National Marine Fisheries Service, the National Environmental Satellite, Data, and Information Administration, and the Institute of Oceanic and Atmospheric Research. A cooperation agreement between the China Meteorological Administration (CMA) and NOAA was signed in 2013 and allows for the public distribution of China meteorological data for the continental region, which is released every three hours. There are more than 300 CMA-NOAA stations in Mainland China, and 1 CMA-NOAA station in Xi'an, see the distribution in Figure 1. PRS (pressure), RH (relative humidity), TEM (temperature), WD (wind direction), and WS (wind speed) are the meteorological factors chosen [22,25].

### 2.2.4. Data Preprocessing and Integration

The pertinent data from various sources must be preprocessed and combined in order to verify the uniformity of the samples. Firstly, the function of the mask algorithm (FMask) provided by GEE is employed to remove clouds, and a circular buffer zone with a radius of 15 m is established with each CNEMC monitoring station as the center. The reflectivity of each band of the Landsat 8 image in the circular buffer zone is extracted. The mean value of the band reflectivity in the circular buffer zone is calculated, and it is assigned to the corresponding CNEMC monitoring station to realize the fusion of Landsat 8 data and $PM_{2.5}$ concentrations. Through the statistical analysis of the distances between the CNEMC and the CMA-NOAA stations using NAA (near analysis algorithm), the nearest CMA-NOAA station to the CNEMC station can be identified, and thus the meteorological data of the CMA-NOAA station are treated as the meteorological attributes of the CNEMC station.

In addition, the meteorological data with incomplete data on atmospheric pressure, temperature, humidity, etc., the $PM_{2.5}$ data with abnormal pollutant concentrations, and the satellite data heavily affected by clouds (BQA $\neq$ 2720) were filtered to enhance the quality of the sample data for machine learning using the tools provided by GEE. A total of 974 samples were obtained.

### 2.3. Methodology

The research methodology employed in this article can be divided into six main stages: (1) collection and fusion of multi-source data, including satellite remote sensing data, PM$_{2.5}$ concentration data, meteorological data, and spatio-temporal data; (2) modeling based on support vector regression (SVR), the multi-layer back propagation network (MLBPN), and random forest (RF); (3) selection of the optimal model for estimating the PM$_{2.5}$ surface concentration in Xi'an according to the evaluation indices; (4) correction of PM$_{2.5}$ surface concentration values using Kriging interpolation to obtain a continuous spatial distribution of PM$_{2.5}$ surface concentrations; (5) superposition of multiple estimation results; and (6) plotting PM$_{2.5}$ concentration contours to analyze the regional representativeness of monitoring stations in Xi'an. The technology roadmap is shown in Figure 2.

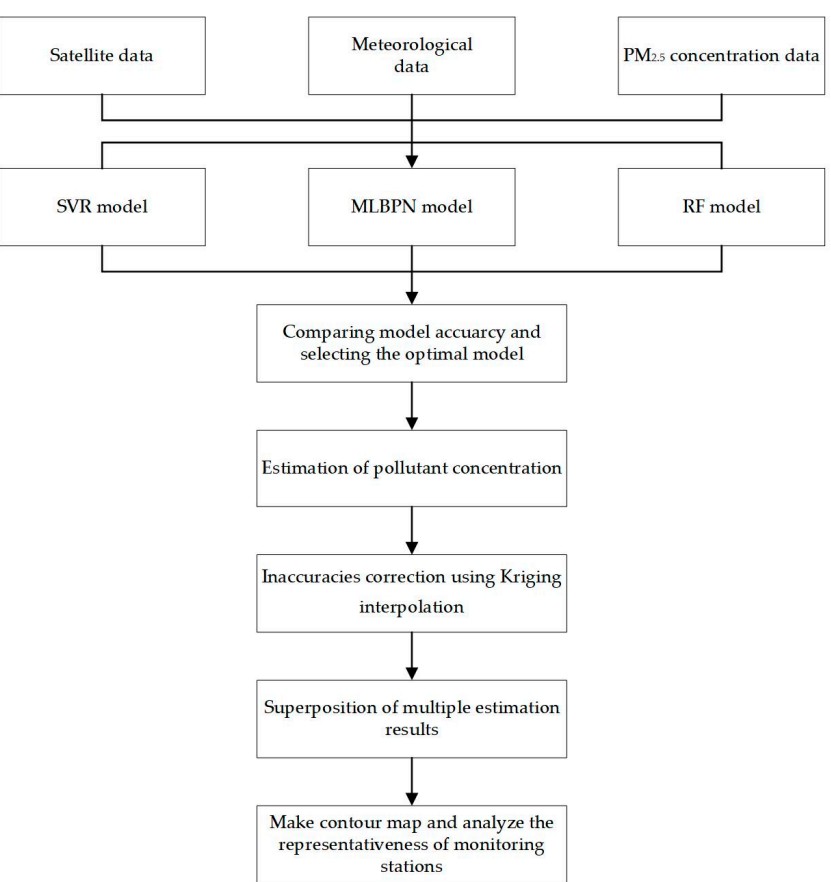

**Figure 2.** Technology roadmap.

### 2.3.1. Multi-Layer Back Propagation Neural Network (MLBPN)-Based PM$_{2.5}$ Estimation Method

The MLBPN constructed in this article consists of an input layer, multiple hidden layers, and an output layer [26]. The units in each layer are connected to all units in the neighboring layers, but no connection exists between units in the same layer. In this project, the input layer is the meteorological data of atmospheric pressure, wind, wind direction, temperature, relative humidity, and satellite remote sensing data, and the output layer is the pollutant concentration estimation value.

In Figure 3, $X_1$, $X_2$, ... , $X_m$ represent the input value of this neural network, and $Y$ represents the output value of this neural network, $s_l$ represents the number of neurons in the *l*th hidden layer, and $w_{ij}^l$ represents the connection weight of the *j*-th neuron in the *l-1*th

hidden layer to the *i*-th neuron in the lth hidden layer, $b_i^l$, and then denotes the bias of the *i*-th neuron in the *l*-th hidden layer, satisfying Equations (1) and (2).

$$b_i^l = f\left(net_i^l\right) \tag{1}$$

$$net_i^l = \sum_{j=1}^{s_{l-1}} w_{ij}^l h_j^{l-1} + b_i^l \tag{2}$$

where $net_i^l$ is the input value of the *i*-th neuron in the *l*-th layer and $f(\cdot)$ denotes the activation function.

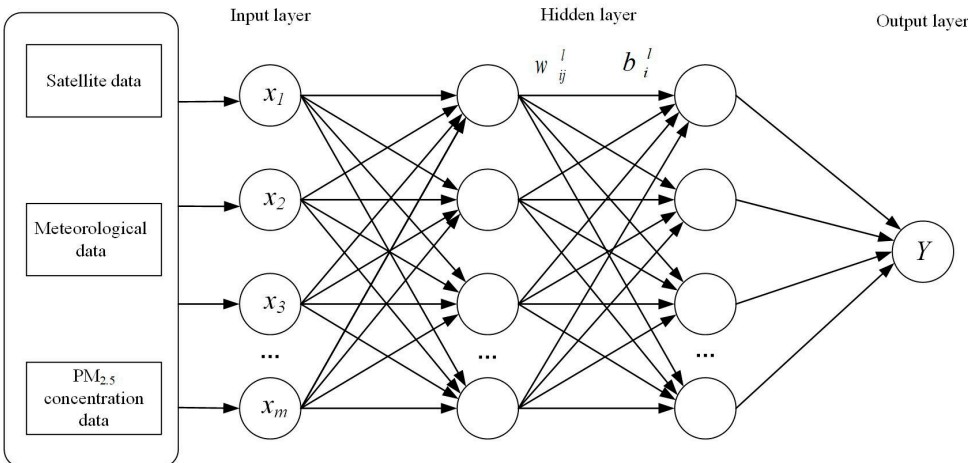

**Figure 3.** Structure diagram of the multi-layer back propagation neural network (MLBPN)-based PM$_{2.5}$ estimation method.

2.3.2. Support Vector Regression (SVR)-Based PM$_{2.5}$ Estimation Method

Support vector machine (SVM) is a novel machine learning technique developed based on statistical learning theory [27,28]. By introducing an insensitive loss function, it can be used for curve fitting to create support vector regression (SVR).

The SVR-based PM$_{2.5}$ estimation model (Figure 4) changes the traditional empirical risk minimization principle and minimizes the expected risk by minimizing the confidence interval as the optimization objective. To calculate the PM$_{2.5}$ concentration values from satellite remote sensing images, the model uses a non-linear function to map the sample points in the low-dimensional space to the high-dimensional space and constructs a linear regression function in the high-dimensional space [28]. The penalty coefficient C and the kernel parameter gamma are selected by iterative selection of the best. The accuracy of support vector regression depends mainly on C and gamma, which are two important model parameters.

In addition, in the process of solving the optimal function using the support vector regression model, an appropriate kernel function needs to be selected to replace the vector inner product in the high-dimensional space, and the commonly used kernel functions are polynomial kernel function, linear kernel function, and radial basis function (RBF). Since the RBF has better generalization ability and can realize nonlinear mapping, the RBF is chosen in this article with the following equation [29].

$$k(x, x_i) = \exp\left\{ -\frac{||x - x_i||^2}{2\sigma^2} \right\} \tag{3}$$

where $\sigma$ is the kernel function, $x$ is the predictor variable, and $x_i$ is the sample factor variable.

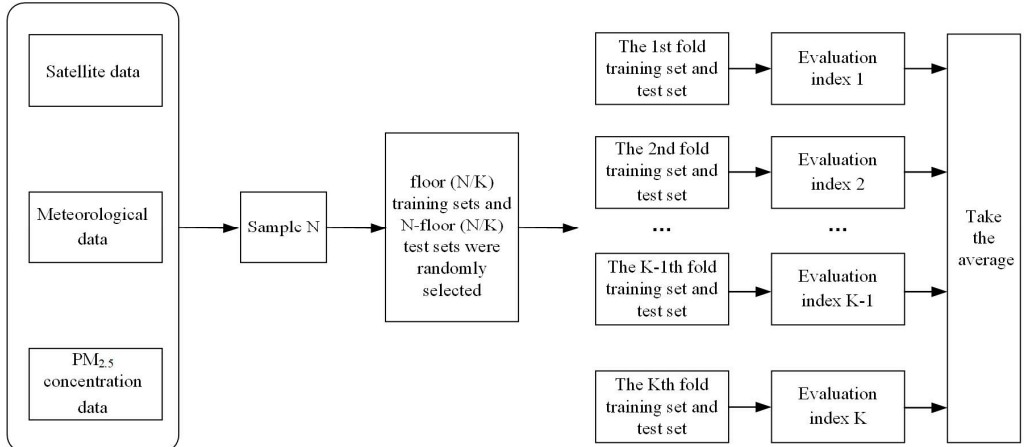

**Figure 4.** Structure diagram of the support vector regression (SVR)-based PM$_{2.5}$ estimation method.

### 2.3.3. Random Forest (RF)-Based PM$_{2.5}$ Estimation Method

The random forest (RF) [30] method is an advanced machine learning technique that optimizes the decision tree method. The RF-based PM$_{2.5}$ estimation model (Figure 5) consists of multiple decision trees, each of which achieve a certain level of accuracy. The overall accuracy is determined by averaging the results of multiple decision trees using the majority voting method. Each decision tree in the random forest model is a binary tree, which follows the top-down recursive splitting growth rule. The nodes in each tree split based on the minimum impurity principle to achieve better accuracy.

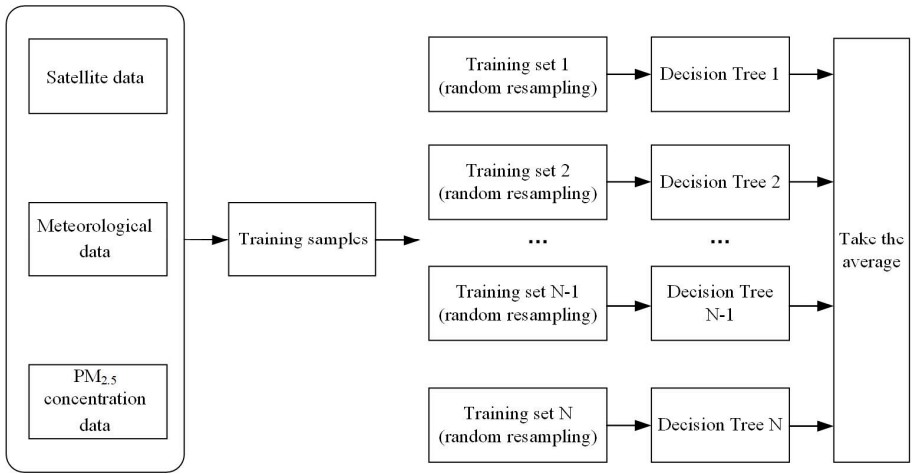

**Figure 5.** Structure diagram of the random forest (RF)-based PM$_{2.5}$ estimation method.

### 2.3.4. Validation

The training and validation datasets were chosen by multiple random sampling using the "leave-out method", which is frequently used in machine learning, in order to assess the accuracy of the PM$_{2.5}$ estimation methods based on the three models. The following three evaluation indicators are also taken into account:

$$ME = \frac{1}{N} \sum_1^N (PM_F - PM_S) \tag{4}$$

$$RMSE = \sqrt{\frac{\sum (PM_F - PM_S)^2}{N}} \tag{5}$$

$$R^2 = \frac{cov(PM_F - PM_S)}{\sqrt{var[PM_F]var[PM_S]}} \tag{6}$$

where $PM_S$ is the estimated PM$_{2.5}$ value; $PM_F$ is the measured PM$_{2.5}$ value; and $N$ is the number of samples.

When selecting the best estimation technique, the fundamental performance of the model is typically evaluated using the mean error (ME), the root means square error (RMSE), and the coefficient of determination ($R^2$). The guiding principle is that "the larger $R^2$, smaller ME and RMSE suggest that the estimated value of the estimation model is closer to the actual value."

### 2.3.5. Error Correction

Due to the inherent errors in the estimated results and the limited nature of remote sensing image data, the superimposition of remote sensing images can only reflect the average value of PM$_{2.5}$ at a few time points, instead of the annual average value. Therefore, it is necessary to correct errors in the estimated results.

Interpolation methods can be divided into deterministic interpolation and geostatistical interpolation according to their mathematical nature. Inverse distance weighting (IDW) is an interpolation method that uses a mathematical function to create an estimated surface. However, the ground statistical interpolation takes into account the autocorrelation between points, and the surface data are obtained according to the statistical characteristics. The commonly used method is Kriging interpolation.

As a random interpolation method based on the generic least squares algorithm and employing the variance map as a weight function, Kriging interpolation, also known as spatial local interpolation, can be used to estimate the distribution of any point data on the surface of the globe. The Kriging method is a true value estimator, and its research can be traced back to the 1950s; the algorithm prototype is the ordinary Kriging method. At present, the ordinary Kriging method is one of the most commonly used methods in Kriging interpolation. According to the principle of the Kriging method, the value of the estimated random field at the sample point is consistent with the corresponding actual measured value. The advantage of this property is that the estimation of the Kriging method is always close to the measured value, not too far away from the actual situation, but the estimated value is always between the measured values and cannot simulate drastic changes. This provides an idea for estimating the error value.

In the process of using IDW, it is required that the monitoring points must be dense enough to cover the range of the estimation surface. The denser the monitoring points, the more accurate the estimation results. The values of monitoring points cannot have large and irregular differences in the estimation range, otherwise, the estimation results are likely to be untrue. Lloyd et al. (2004) conducted spatial interpolation analysis on the concentration of atmospheric pollutants in the UK, using the ordinary Kriging method and the IDW method, and the results showed that the interpolation effect of the ordinary Kriging method is better than that of the IDW method [31].

Considering that the estimated PM$_{2.5}$ concentration could be different from the actual value, it is necessary to correct the errors in the estimated results. Therefore, a method of Kriging interpolation has been proposed to enhance the estimation accuracy of PM$_{2.5}$ concentration. The specific process of error-correction method is depicted in Figure 6: firstly, calculate the difference (ref. δ in Figure 7) between the actual value and the estimated value of the PM$_{2.5}$ concentration at each station (ref. S in Figure 7), then, the difference values are employed as the input data for the interpolation in the study area by the Kriging method; after that, one image depicting the spatial distribution of the differences (δ) can be achieved, which will be superimposed to the estimated image of the PM$_{2.5}$ concentration and thus the estimation errors could be corrected. As depicted by Figure 7, the leave-one-out method is employed to validate the effectiveness of Kriging interpolation, where one station is randomly selected for verification (ref. the red point S$_6$ in Figure 7) and the difference

values of several other stations are used for interpolation. By comparing the absolute value of error before ($\delta_6$) and after error correction ($\delta_6{}'$), the feasibility will be verified.

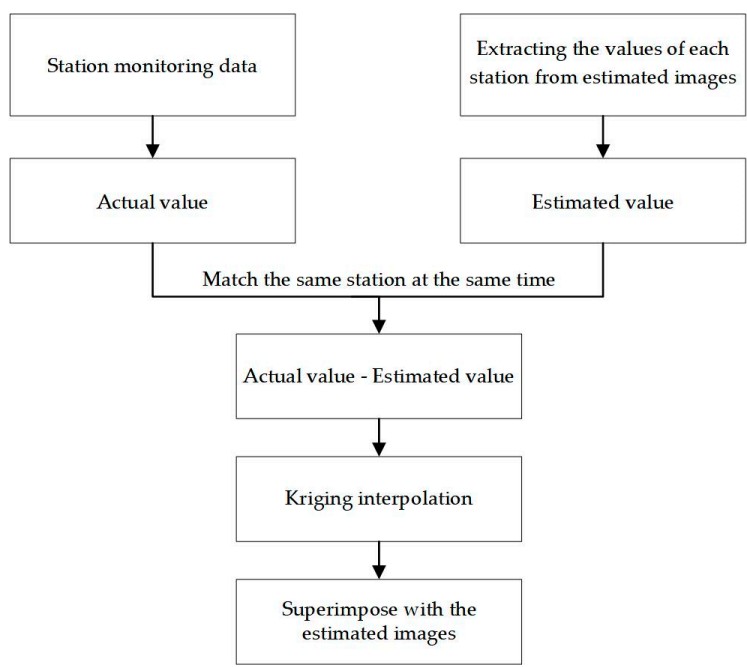

**Figure 6.** The method of error correction.

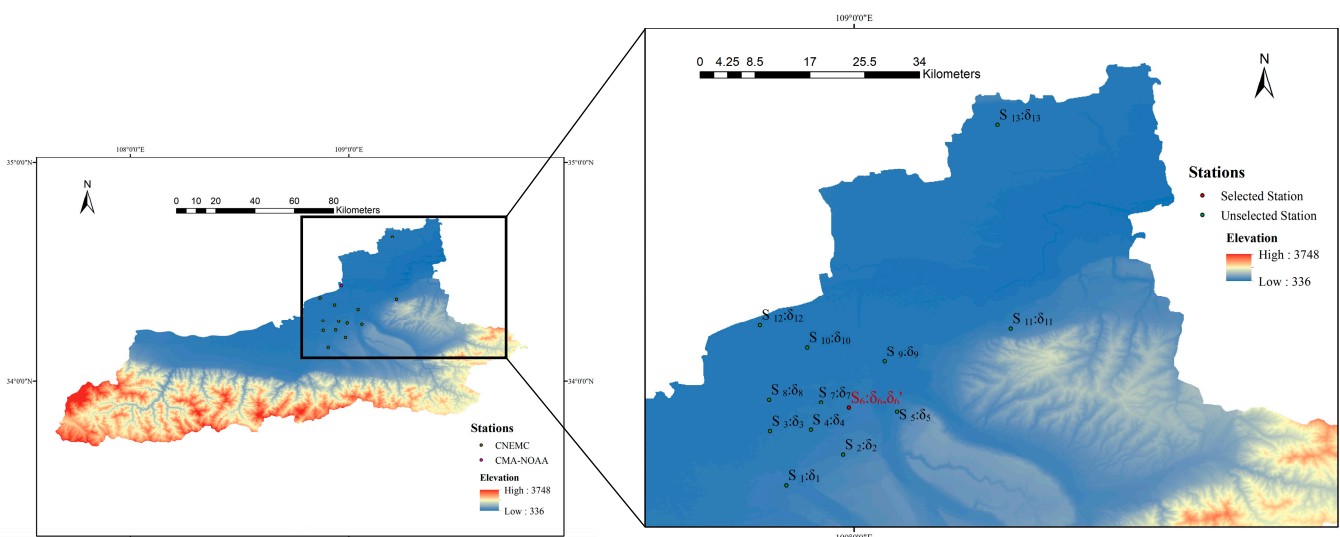

**Figure 7.** The leave-one-out method was employed to prove the effectiveness of Kriging interpolation.

### 2.3.6. Superimposition of Multiple Estimated Results

Due to the better data quality in 2018, this article selected the images of this year for further analysis. After correcting the errors of the daily estimated images of the $PM_{2.5}$ concentration in 2018, they were superimposed and averaged to obtain the annual average pollutant distribution image. After extracting the annual average $PM_{2.5}$ concentration value of the estimated image at one station, the difference to the actual annual average monitoring value of the corresponding station can be calculated, which will be taken as the input data to conduct the Kriging interpolation for the error correction in the study area.

### 2.3.7. Representativeness Analysis

At present, there are 13 national ground monitoring stations in Xi'an, mainly located in Weiyang District, Xincheng District, Beilin District, Yanta District, and other areas, with fewer suburban and field environmental monitoring stations. Because of Xi'an's unique geographical location, it is difficult to set up monitoring stations in the southern Qinling Mountains, making pollutant monitoring work in the Qinling Mountains more difficult. It is currently proposed to establish a surface environmental monitoring network through collaborative prevention and control and to judge the environmental data of southern Xi'an using monitoring data from other cities. However, the terrain in the west of Xi'an is flat, and many enterprises have built factories here to discharge sewage. What is more, there are relatively few monitoring stations in this location, and the other four cities in Guanzhong have not set up environmental monitoring stations in the nearby borders, which has caused certain difficulties in pollutant monitoring.

This project visualizes the estimated value and makes a contour map of pollutants in Xi'an. What is more, a method of station representativeness analysis has been proposed, as shown in Figure 8. By comparing the average value of the estimated pollutant concentration in the region with the average value of the pollutant concentration of 13 stations separately, a quantitative evaluation of regional representativeness can be provided. If the difference between the regional average value and the station average value is small, it can be considered that the representativeness of the monitoring station is good. If the gap is large, the reason should be analyzed in combination with the location of the station.

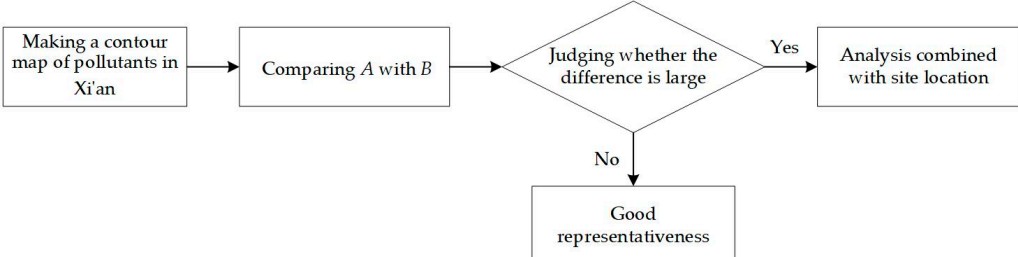

**Figure 8.** Structure diagram of representativeness analysis. In the second step, *A* represents the average value of the estimated pollutant concentration in the region, and *B* represents the average value of the pollutant concentration of each station.

## 3. Results

### 3.1. Analysis of Estimation Accuracy

Band 1, Band 3, Band 7, the NDVI of the Landsat 8 satellite, and meteorological factors obtained from CMA-NOAA were used as input parameters to MLBPN, RF, and SVR to obtain the $PM_{2.5}$ estimation model. Table 2 shows the accuracy evaluation of different models. According to the principle of "the larger $R^2$, smaller ME and RMSE suggest that the estimated value of the estimation model is closer to the actual value", and as shown in the scatter plot of correlation between estimated and actual values (Figure 9), it can be seen that the MLBPN-based $PM_{2.5}$ estimation model performs better than both of the RF-based and the SVR-based $PM_{2.5}$-estimation models.

**Table 2.** Accuracy evaluation of different models.

| Model | $R^2$ | ME/μg·m$^{-3}$ | RMSE/μg·m$^{-3}$ |
|---|---|---|---|
| MLBPN | 0.90 | −1.01 | 20.57 |
| RF | 0.82 | 4.02 | 23.04 |
| SVR | 0.79 | −2.15 | 27.60 |

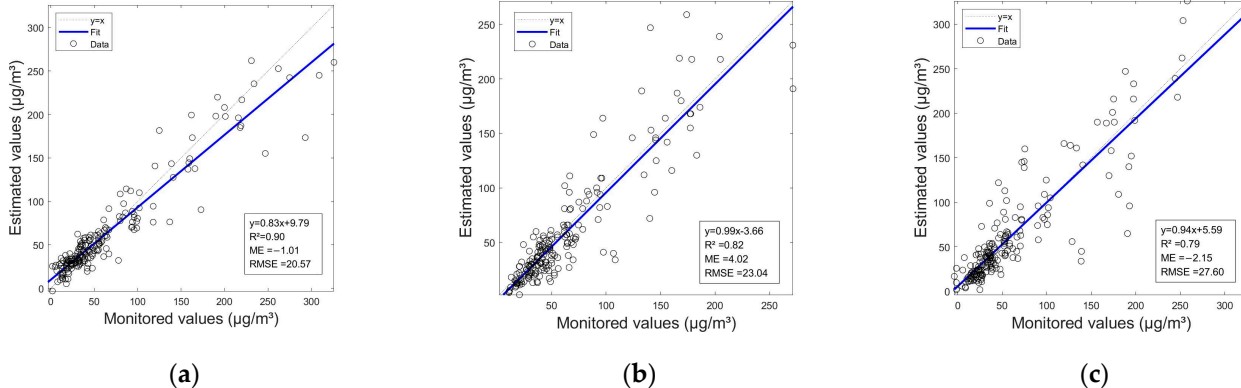

**Figure 9.** Correlation between the estimated and monitored values of PM$_{2.5}$ concentrations. (**a**) MLBPN. (**b**) RF. (**c**) SVR.

Figure 10 compares the agreement between the estimated PM$_{2.5}$ concentrations and the actual monitoring values of the three models. The blue lines in the figure represent the actual monitored values, and the red circles represent the estimated values. Overall, when the PM$_{2.5}$ concentration is high, the estimated values have a larger error than the actual values. Meanwhile, the consistency between the estimated and actual monitored values of the MLBPN-based PM$_{2.5}$ estimation model is better than that of the RF-based PM$_{2.5}$ estimation model and the SVR-based PM$_{2.5}$ estimation model. In summary, the MLBPN-based PM$_{2.5}$ estimation model was selected for the next study in this article.

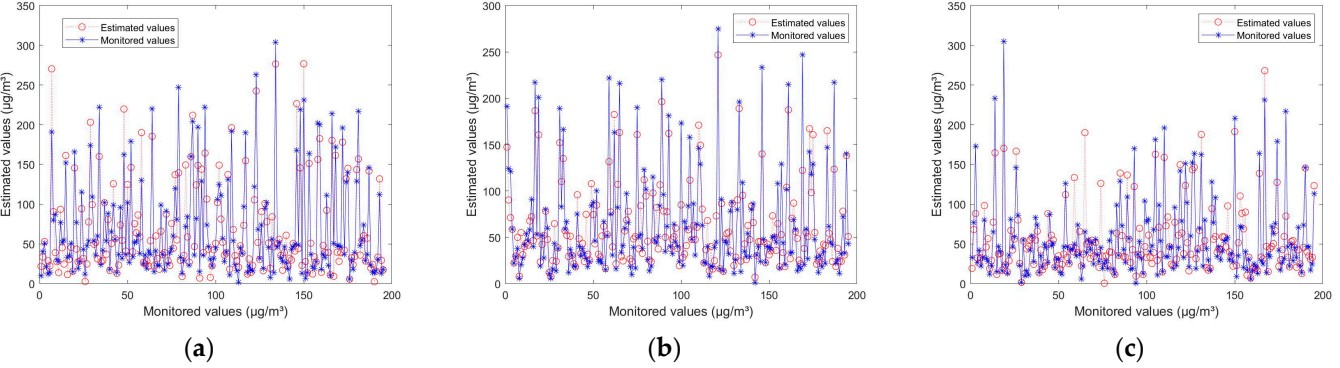

**Figure 10.** Comparison of the estimated and monitored values of PM$_{2.5}$ concentrations. (**a**) MLBPN. (**b**) RF. (**c**) SVR.

### 3.2. Error Correction

Due to the better data quality in 2018, the annual Landsat 8 remote sensing images, corresponding meteorological data, and PM$_{2.5}$ concentrations at monitoring stations in Xi'an for this year were selected to estimate pollutant concentrations with the trained MLBPN-based PM$_{2.5}$ estimation model to obtain annual PM$_{2.5}$ estimation images. After extracting the pollutant concentration values for each station individually, the differences were calculated based on the actual monitoring values for the corresponding stations on that day. The obtained differences were then subjected to Kriging interpolation to estimate the errors for the purpose of error correction. In order to verify the feasibility of the Kriging interpolation method, the 1463A station was randomly selected to perform Kriging interpolation with the difference values of other stations. Some study cases were randomly picked out as shown in Table 3; they indicated that the value after error correction is closer to the actual value. By comparing the absolute value of error before and after error correction of 1463A, as shown in Table 4, the absolute value of mean error before error correction is 18.78 µg·m$^{-3}$, and the absolute value of mean error after error correction is

2.08 μg·m$^{-3}$, indicating that using Kriging interpolation can improve the accuracy, and the error correction method proposed in this article is also feasible to other areas.

**Table 3.** Comparison before and after error correction of 1463A.

| Date | Actual Value/μg·m$^{-3}$ | Estimated Value/μg·m$^{-3}$ | \|Actual Value -Estimated Value\| /μg·m$^{-3}$ | Interpolation Value/μg·m$^{-3}$ | The Value After Error Correction /μg·m$^{-3}$ |
|---|---|---|---|---|---|
| 20180215 | 84.00 | 106.97 | 22.97 | 25.30 | 81.67 |
| 20180303 | 189.00 | 153.74 | 35.26 | 35.42 | 189.16 |
| 20180522 | 93.00 | 70.66 | 22.34 | 21.75 | 92.41 |
| 20180623 | 48.00 | 33.94 | 14.96 | 12.16 | 46.10 |
| 20180725 | 13.00 | 19.03 | 12.45 | 9.03 | 10.00 |
| 20180826 | 38.00 | 21.15 | 16.85 | 14.52 | 35.67 |
| 20180911 | 16.00 | 4.67 | 11.32 | 6.16 | 10.83 |
| 20181016 | 43.00 | 57.04 | 14.05 | 12.92 | 44.12 |

**Table 4.** Comparison of the absolute value of error before and after error correction of 1463A.

| Date | The Absolute Value of Error before Error Correction/μg·m$^{-3}$ | The Absolute Value of Error after Error Correction/μg·m$^{-3}$ |
|---|---|---|
| 20180215 | 22.97 | 2.33 |
| 20180303 | 35.26 | 0.16 |
| 20180522 | 22.34 | 0.59 |
| 20180623 | 14.96 | 1.90 |
| 20180725 | 12.45 | 3.00 |
| 20180826 | 16.85 | 2.33 |
| 20180911 | 11.32 | 5.17 |
| 20181016 | 14.05 | 1.12 |
| The average | 18.78 | 2.08 |

To make more precise evaluations, the study case was randomly picked out in the test area, as shown in Figure 11. In addition, Table 5 shows the PM$_{2.5}$ concentrations estimated by Landsat 8 in Xi'an at UTC 3:19 on 14 January 2018, and the results were obtained by correcting the errors through Kriging interpolation. By comparing the corrected pollutant concentrations with the originally estimated pollutant concentrations, it shows that the use of Kriging interpolation can effectively correct the experimental errors to a certain degree.

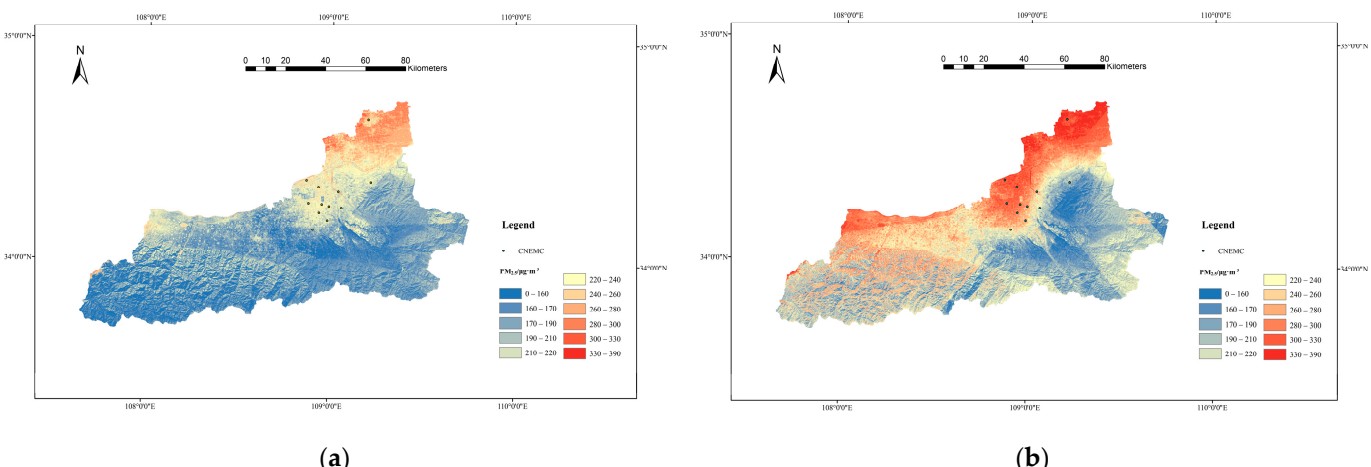

**Figure 11.** Comparison before and after error correction in Xi'an at UTC 3:19 on 14 January 2018. (**a**) Estimated value. (**b**) The value after error correction.

**Table 5.** Landsat 8-estimated $PM_{2.5}$ concentration and correction in Xi'an at UTC 3:19 on 14 January 2018.

| Station ID | Actual Value/$\mu g \cdot m^{-3}$ | Estimated Value/$\mu g \cdot m^{-3}$ | \| Actual Value-Estimated Value \| /$\mu g \cdot m^{-3}$ | The Value after Error Correction /$\mu g \cdot m^{-3}$ |
|---|---|---|---|---|
| 1462A | 279.00 | 210.19 | 68.81 | 272.76 |
| 1463A | 239.00 | 188.26 | 50.74 | 232.63 |
| 1464A | 231.00 | 230.92 | 0.08 | 237.08 |
| 1465A | 275.00 | 207.26 | 67.74 | 268.68 |
| 1466A | 262.00 | 221.73 | 40.27 | 286.50 |
| 1468A | 304.00 | 224.55 | 79.45 | 291.68 |
| 1469A | 218.00 | 172.68 | 45.32 | 222.44 |
| 1470A | 309.00 | 233.02 | 75.98 | 307.55 |
| 1471A | 191.00 | 221.54 | 30.54 | 193.12 |
| 1472A | 326.00 | 248.69 | 77.31 | 311.79 |
| 1473A | 284.00 | 233.71 | 50.29 | 276.47 |
| 1474A | 234.00 | 201.06 | 32.94 | 231.52 |

The annual average $PM_{2.5}$ estimation map for 2018 was created in order to more intuitively evaluate the representativeness of the monitoring stations in Xi'an, and the experimental error was further corrected using Kriging interpolation. After extracting the annual average $PM_{2.5}$ concentration values of pollutants at the stations, the differences were made with the actual annual average monitoring values at the corresponding stations. The obtained difference values were employed as the input data of the Kriging interpolation to correct the errors, as shown in Figure 12. As can be seen from Table 6, for the annual average $PM_{2.5}$ in Xi'an, the error between the corrected and actual values is reduced after correction. Through calculation, the corrected value has an RMSE of 2.11 $\mu g \cdot m^{-3}$ and an ME of 0.12 $\mu g \cdot m^{-3}$ with the actual value, which is much more accurate.

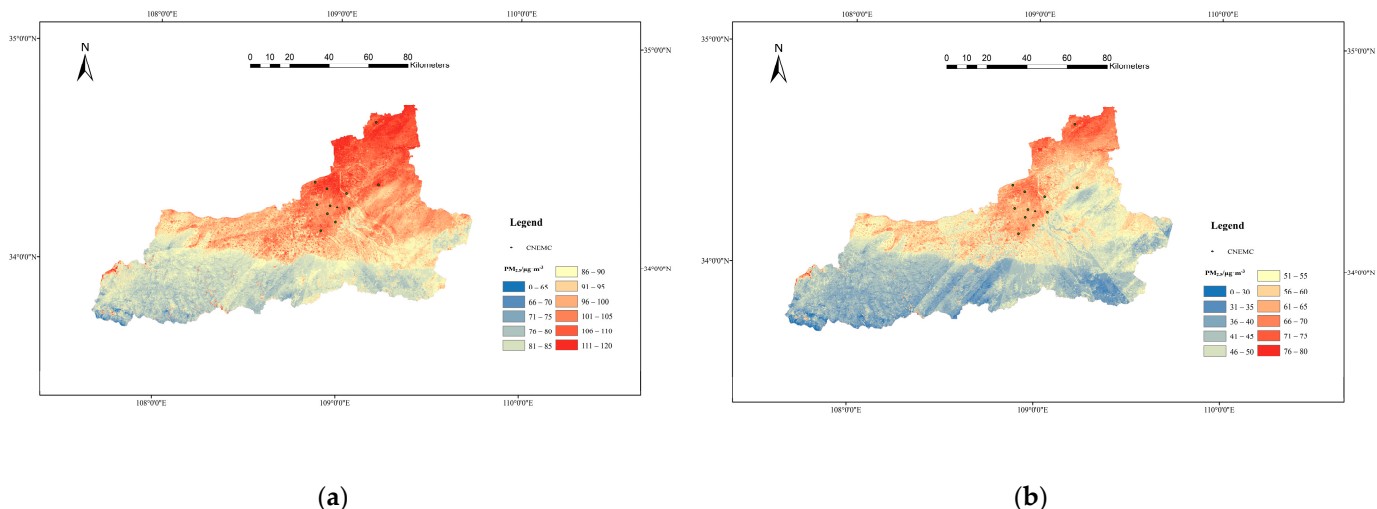

(**a**)                                                                                          (**b**)

**Figure 12.** Comparison before and after error correction of annual pollutant concentration distribution in Xi'an in 2018. (**a**) Estimated annual value. (**b**) The annual value after error correction.

### 3.3. Representativeness Analysis

Based on the spatial distribution of the revised annual average $PM_{2.5}$ estimation concentration of Xi'an in 2018 (ref. Figure 12b), the pollutant contour map was drawn with a contour spacing of 10 $\mu g \cdot m^{-3}$ (ref. Figure 13). As depicted in Figure 13, the $PM_{2.5}$ concentration in northwest region is significantly higher than southwest region, which is consistent with the characteristics of population and industrial distributions in the study area.

**Table 6.** Landsat 8-estimated annual average PM$_{2.5}$ concentration and correction in Xi'an in 2018.

| Station ID | Actual Value/µg·m$^{-3}$ | Estimated Value/µg·m$^{-3}$ | Actual Value-Estimated Value \| /µg·m$^{-3}$ | The Value after Error Correction/µg·m$^{-3}$ |
|---|---|---|---|---|
| 1462A | 64.60 | 100.24 | 35.64 | 61.00 |
| 1463A | 54.77 | 97.29 | 42.52 | 57.79 |
| 1464A | 63.65 | 101.78 | 38.13 | 60.56 |
| 1465A | 61.29 | 100.44 | 39.15 | 61.91 |
| 1466A | 63.04 | 99.07 | 36.03 | 60.02 |
| 1467A | 60.24 | 98.84 | 38.60 | 59.88 |
| 1468A | 63.24 | 105.55 | 42.31 | 63.71 |
| 1469A | 58.60 | 100.55 | 41.95 | 59.85 |
| 1470A | 73.33 | 110.36 | 37.03 | 71.77 |
| 1471A | 54.16 | 107.10 | 52.94 | 57.02 |
| 1472A | 61.53 | 106.63 | 45.11 | 63.44 |
| 1473A | 60.05 | 98.42 | 38.37 | 59.45 |
| 1474A | 58.73 | 102.26 | 43.52 | 59.34 |

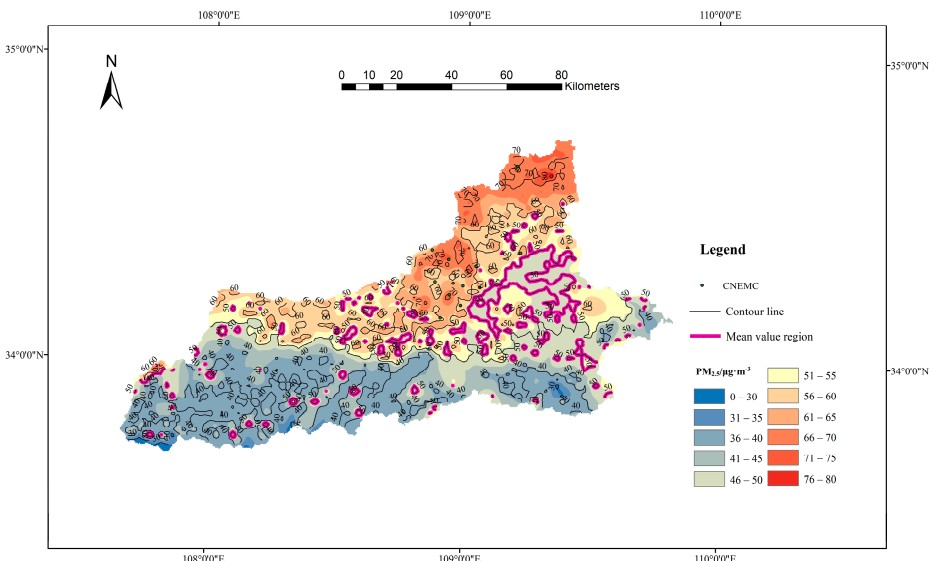

**Figure 13.** The contour lines of the estimated annual average PM$_{2.5}$ concentration and the medium area of Xi'an in 2018.

The annual average PM$_{2.5}$ concentration monitored at the station is 61.33 µg·m$^{-3}$, and the estimated station annual mean is 61.21 µg·m$^{-3}$ (after error correction), which indicates that the estimation method proposed in this article has good arithmetic results. In contrast, the annual average PM$_{2.5}$ estimation concentration in the whole region of Xi'an is 49.89 µg·m$^{-3}$; by calculating the difference with the actual annual average value of each station, it can be seen from Table 7 that the difference of the 1471A station is 7.13 µg·m$^{-3}$, which is the smallest and indicates that the representativeness of the station is relatively good, and the regional annual average value of PM$_{2.5}$ in Xi'an is overestimated by 14.29%. Followed by 1463A, the difference is 7.90 µg·m$^{-3}$, overestimating the regional annual average value of PM$_{2.5}$ in Xi'an by 15.83%. Furthermore, 1474A, 1473A, 1469A, and 1467A have differences between 9.45 µg·m$^{-3}$ and 9.99 µg·m$^{-3}$, which overestimate the regional annual average value of PM$_{2.5}$ in Xi'an from 18.94% to 20.02%. Then, 1466A, 1464A, 1462A, 1465A, 1472A, and 1468A have differences between 10.13 µg·m$^{-3}$ and 13.82 µg·m$^{-3}$, which overestimate the regional annual average value of PM$_{2.5}$ in Xi'an from 20.30% to 27.70%. The 1470A station has the largest difference of 21.88 µg·m$^{-3}$, which overestimates the regional annual average value of PM$_{2.5}$ in Xi'an by 43.86%.

**Table 7.** The difference between the actual station annual average and the annual average of estimated PM$_{2.5}$ concentration in the whole region.

| Station ID | The Station Annual Average/μg·m$^{-3}$ | The Difference [1]/μg·m$^{-3}$ | Percentage of Error |
|:---:|:---:|:---:|:---:|
| 1462A | 61.00 | 11.11 | 22.27% |
| 1463A | 57.79 | 7.90 | 15.83% |
| 1464A | 60.56 | 10.67 | 21.39% |
| 1465A | 61.91 | 12.02 | 24.09% |
| 1466A | 60.02 | 10.13 | 20.30% |
| 1467A | 59.88 | 9.99 | 20.02% |
| 1468A | 63.71 | 13.82 | 27.70% |
| 1469A | 59.85 | 9.96 | 19.96% |
| 1470A | 71.77 | 21.88 | 43.86% |
| 1471A | 57.02 | 7.13 | 14.29% |
| 1472A | 63.44 | 13.55 | 27.16% |
| 1473A | 59.45 | 9.56 | 19.16% |
| 1474A | 59.34 | 9.45 | 18.94% |

[1] The difference = the actual station annual average. The annual average of estimated PM$_{2.5}$ concentration in the whole region (49.89 μg·m$^{-3}$).

Monitoring stations need to have good representativeness, which can objectively reflect the level and variation of ambient air quality in a certain spatial range and objectively evaluate the urban and regional ambient air conditions and the impact of pollution sources on ambient air quality, so as to meet the needs of providing health guidance for the public. Through the analysis, it can be concluded that the average value of the estimated pollutant concentration in the region is quite different from the annual average value of each station. On the whole, the regional annual average value of PM$_{2.5}$ in Xi'an will be overestimated. It can be seen from the map that the monitoring stations in Xi'an are concentrated in the main urban area and distributed densely. Due to the influence of the Qinling Mountains, the PM$_{2.5}$ concentration in Xi'an is quite different between the north and the south. The monitoring stations concentrated in the main urban area find it difficult to reflect the PM$_{2.5}$ pollution level in Xi'an, and the representativeness is not good enough. In order to provide a scientific basis for the station selection of monitoring stations, taking the annual average PM$_{2.5}$ estimation concentration in the whole region of Xi'an as the standard, the area with the contour value of 50 μg·m$^{-3}$ was selected. This area is the mean value region of pollutants and has good representativeness. Therefore, the mean value region was selected as a representative station location for reference, as shown in Figure 13.

## 4. Discussion

In this article, we collected Landsat 8 remote sensing image data of Xi'an for five years from 2017 to 2021 on the GEE platform, integrated air quality data from CNEMC and meteorological data from CMA-NOAA, and constructed PM$_{2.5}$ estimation models based on MLBPN, RF, and SVR. The R$^2$ of PM$_{2.5}$ estimation models based on MLBPN, RF, and SVR are 0.90, 0.82, and 0.79, based on ME, they are $-1.01$ μg·m$^{-3}$, 4.02 μg·m$^{-3}$, and $-2.15$ μg·m$^{-3}$, and based on RMSE, they are 20.57 μg·m$^{-3}$, 23.04 μg·m$^{-3}$, and 27.60 μg·m$^{-3}$, respectively. According to the evaluation principles, the MLBPN-based PM$_{2.5}$ estimation model was selected as the optimal model. The PM$_{2.5}$ concentration distribution map obtained in this article has a spatial resolution of 30 m, which is more accurate than other PM$_{2.5}$ concentration distribution maps obtained based on satellite images. Moreover, the data selected in this article are all open source, which makes the model highly versatile. In this article, PM$_{2.5}$ was selected as a representative pollutant for the study, but the model is also applicable to PM$_{10}$, O$_3$ and other pollutants.

This article proposes a novel method to correct errors in PM$_{2.5}$ estimation using Kriging interpolation, which significantly improves the accuracy of the estimation model. Meanwhile, due to the better data quality in 2018, the images in this year were selected for

further analysis. From the contour maps of pollutants and the comparison of the actual annual average $PM_{2.5}$ concentrations monitored at the stations, the estimation station annual average value, and the annual average $PM_{2.5}$ estimation concentrations in Xi'an, it is clear that the estimation model created in this article has accurate results and strong application value.

However, the article also highlights the inadequate representativeness of monitoring stations in Xi'an, and they will overestimate the regional annual average value of $PM_{2.5}$ in Xi'an. In this article, the annual average $PM_{2.5}$ estimation concentration in the whole region of Xi'an was used as the standard to select the median pollutant area. The mean value region was selected as the better representative station location. Although we selected images in 2018 for further analysis, the method is equally applicable to successive years of images. In summary, this article can provide decision support for the site selection of well-represented monitoring stations.

## 5. Conclusions

In this article, the multispectral data of Landsat 8 data, air quality data, and meteorological data from ground monitoring stations were integrated together and imported to different $PM_{2.5}$-estimation models based on the multi-layer back propagation neural network (MLBPN), support vector regression (SVR), and random forest (RF), respectively. As measured by the evaluation indices of $R^2$, RMSE, and ME, the estimation model based on the MLBPN produced the best $PM_{2.5}$ estimation results and was thus employed for the regional representativeness analysis in the study area of Xi'an, Shaanxi, China. The annual average $PM_{2.5}$ concentration in the study area was depicted after error correction using Kriging interpolation, which can then be used to evaluate and analyze the representativeness of Xi'an's monitoring stations.

This article presents a comprehensive approach to addressing the problem of the under-representativeness of monitoring stations in urban areas of China, particularly for $PM_{2.5}$. The proposed MLBPN-based $PM_{2.5}$ estimation model, with the additional Kriging interpolation method for error correction, has demonstrated high accuracy in comparison to other estimation models. Moreover, contour maps based on the estimated annual average $PM_{2.5}$ values are also useful for judging the regional representativeness of the environmental monitoring stations in the study area. By calculating the difference between the actual station annual average and the annual average of estimated $PM_{2.5}$ concentration in the whole region, it can be found that the regional annual average value of $PM_{2.5}$ in Xi'an is overestimated, indicating the inadequate representativeness of the stations. Based on the annual average $PM_{2.5}$ estimation concentration in the whole region of Xi'an and the median area of pollutants, the authors selected the locations of the stations with better representativeness.

To sum up, the article scientifically evaluates and analyzes the regional representativeness of the stations in Xi'an and provides new methods for the siting of environmental monitoring stations. The model proposed in this article has strong versatility, can obtain continuous pollutant distribution maps, and provides decision support for the site selection of new stations. The well-represented stations can objectively reflect the ambient air quality level within a specific spatial range, objectively evaluate the regional ambient air conditions, and meet public health guidance needs for ambient air conditions. It must be mentioned that although only the $PM_{2.5}$ concentration distribution was analyzed in this article, a similar approach can be employed to analyze some other air pollutants in the future, such as $PM_{10}$ and $O_3$, so as to evaluate the representativeness of air quality monitoring stations more comprehensively.

**Author Contributions:** Conceptualization, R.L. and M.Z.; methodology, M.Z.; R.L., M.Z. and G.M. collected the data and completed the experiments; M.Z. initiated the overall research question, and M.Z. found funding for this project; R.L. drafted the paper; M.Z. and G.M. critically read and revised the draft. All authors have read and agreed to the published version of the manuscript.

**Funding:** This research was funded by the Key Research and Development Program of Shaanxi Province (China) under grant 2020SF-434 and in part by the National Natural Science Foundation of China under grant 41871315.

**Data Availability Statement:** Not applicable.

**Acknowledgments:** This work was supported by the Strategic Priority Research Program of the Chinese Academy of Sciences under Grant XDB40020200.

**Conflicts of Interest:** The authors declare no conflict of interest.

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
