# Peer review of "Regional Representativeness Analysis of Ground-Monitoring PM2.5 Concentration Based on Satellite Remote Sensing Imagery and Machine Learning Techniques"

_remotesensing, doi:10.3390/rs15123040_

Round 1
Reviewer 1 Report
This manuscript applies three machine learning methods to retrieve PM2.5 data from the Xi'an area, and introduces the use of Kriging interpolation for error correction. The results demonstrate that the adjusted mapping results seem to significantly improve the accuracy of retrieval. However, I have some major and minor concerns as outlined below:
Major concerns:
1、Line 128: Please clarify why you have chosen only three bands (B1, B3, B7) in Landsat. Additionally, Table 4 suggests that you have also used the band (B10, B11) corresponding to the TIRS sensor, which appears inconsistent. Please address this issue.
2、Line 156: The section on Data Preprocessing and Integration seems overly simplistic and could easily cause confusion. For instance, meteorological data are site-specific. Could you elaborate on the method used to match this with overall spatial satellite data? Have you used nearest neighbor interpolation or a different technique? Furthermore, could you specify the method used for data deletion? Is it based on three standard deviations, or something else? More detail in this section would be beneficial.
3、There appears to be an error in Figure 4 as it includes two K-1th folds. Please correct this issue.
4、What is the resolution of the error value of the Kriging resampling? Does it adhere to the Landsat resolution?
5、I have particular concerns about whether this approach truly provides a representative spatial representation of the site. The results seem to reflect the deviation of Kriging's method in site interpolation?
Minor concerns:
1、The scale bar unit in the figure should be in kilometers (km).
Author Response
Thanks a lot for taking the time to review our article. Your comments and suggestions are very valuable and helpful for the improvements of this manuscript. According to the comments/suggestions of all of the reviewers, the manuscript has been substantially refined from both technical and linguistic points of view, marked using the “Track Changes” function in the attached word-file. Moreover, the final version after the revision (without the tracked changes and marked by the ‘highlight color’) can be found in the attached PDF-file.
By the way, please see below our point-to-point responses to your comments.
Point 1: Line 128: Please clarify why you have chosen only three bands (B1, B3, B7) in Landsat. Additionally, Table 4 suggests that you have also used the band (B10, B11) corresponding to the TIRS sensor, which appears inconsistent. Please address this issue.
Response 1: Thank you for pointing them out. According to some literatures published so far, the Band 1, 3, 7 besides the calculated NDVI based on the Bands 4 and 5 from Landsat 8 OLI, are strongly relevant to the PM2.5 concentration. Hence, the Band 1, 3, 7 and NDVI of Landsat 8 have been employed as the primary source data in our research. In the revised manuscript, we have appended the relevant descriptions (please ref. the sentence of “Based on some literatures published so far [22,25], B1 (0.43-0.45 µm), B3 (0.53-0.59 µm), B7 (2.11-2.29 µm) and the calculated NDVI based on the B4 (0.64-0.67 µm) and B5 (0.85-0.88 µm) of Landsat 8 are strongly relevant to the PM2.5 concentrations. Therefore, B1, B3, B7 and the NDVI of Landsat 8 have been selected as the primary input parameters for the PM2.5 concentration estimations.” in the end of section “2.2.1 Satellite Remote Sensing Data”. Hopefully, it could become clearer. We don’t used the band (B10, B11) corresponding to the TIRS sensor, therefore we have deleted the “TIRS” in Table 1.
- Zhang, B.; Zhang, M.; Kang, J.; Hong, D.; Xu, J.; Zhu, X. Estimation of PMx Concentrations from Landsat 8 OLI Images Based on a Multilayer Perceptron Neural Network. Remote Sens. 2019, 11, 646. https://doi.org/10.3390/rs11060646.
- Li, R.; Sun, L. An Improved DDV Method to Retrieve Aerosol Optical Depth for Landsat 8 OLI Image. Remote Sens. Inf. 2017, 32, 120–125. https://doi.org/10.3969/j.issn.1000-3177.2017.01.021.
Point 2: Line 156: The section on Data Preprocessing and Integration seems overly simplistic and could easily cause confusion. For instance, meteorological data are site-specific. Could you elaborate on the method used to match this with overall spatial satellite data? Have you used nearest neighbor interpolation or a different technique? Furthermore, could you specify the method used for data deletion? Is it based on three standard deviations, or something else? More detail in this section would be beneficial.
Response 2: Thank you for your insightful suggestion, we have rewritten the section on Data Preprocessing and Integration. See details below:
“Firstly, the function of mask algorithm (FMask) provided by GEE is employed to remove clouds, and a circular buffer zone with a radius of 15 meters is established with each CNEMC monitoring station as the center. The reflectivity of each band of the Landsat 8 image in the circular buffer zone is extracted. Calculate the mean value of the band reflectivity in the circular buffer zone and assign it to the corresponding CNEMC monitoring station to realize the fusion of Landsat 8 data and PM2.5 concentrations. By the statistical analysis of the distances between the CNEMC and the CMA-NOAA stations using NAA (Near Analysis Algorithm), the nearest CMA-NOAA station to the CNEMC station can be identified, and thus the meteorological data of the CMA-NOAA station are treated as the meteorological attributes of the CNEMC station.
In addition, the meteorological data with incomplete data of atmospheric pressure, temperature, humidity, etc., the PM2.5 data with abnormal pollutant concentrations, and the satellite data heavily affected by clouds (BQA≠2720) have been filtered to enhance the quality of sample data for machine learning, using the tools provided by GEE.”
Point 3: There appears to be an error in Figure 4 as it includes two K-1th folds. Please correct this issue.
Response 3: Thank you for pointing out the serious mistake. We have corrected it in Figure 4.
Point 4: What is the resolution of the error value of the Kriging resampling? Does it adhere to the Landsat resolution?
Response 4: Thank you for pointing them out. The resolution of the error value of the Kriging resampling is 30 m, and it adhere to the Landsat resolution.
Point 5: I have particular concerns about whether this approach truly provides a representative spatial representation of the site. The results seem to reflect the deviation of Kriging's method in site interpolation?
Response 5: Thank you for pointing them out. we have improved the description of the following paragraph in the revised manuscript. See details below:
“Considering the estimated PM2.5 concentration could be different from the actual value, it is necessary to correct the errors in the estimated results. Thereby, a method of Kriging interpolation has been proposed to enhance the estimation accuracy of PM2.5 concentration. The specific method is to decrease the difference between the actual value and the estimated value of each station firstly, then after taking the absolute value of the obtained value, use ArcGIS Ordinary Kriging method for interpolation, and finally superimpose with the estimated images, as shown in Figure 6. As depicted by Figure 7, leave-one-out method is employed to prove the effectivity of the Kriging interpolation, where one station is randomly selected for verification (the red station S6) and the difference values of several other stations are used for interpolation. By comparing the absolute value of error before (δ6) and after error correction (δ6’), the feasibility will be verified.” (in the end of the section 2.3.5 Error Correction)
“1463A station was randomly selected to perform Kriging interpolation with the absolute value of the error of other stations. Some study cases were randomly picked out as shown in Table 3, it indicated that the value after error correction is closer to the actual value. By comparing the absolute value of error before and after error correction of 1463A, as shown in Table 4, the absolute value of mean error before error correction is 18.78 µg·m-3, and the absolute value of mean error after error correction is 2.08 µg·m-3, indicating that using Kriging interpolation can improve the accuracy, and the error correction method proposed in this article is also feasible to other areas.”(in the end of the first paragraph of the section 3.2 Error Correction)
Point 6: The scale bar unit in the figure should be in kilometers (km).
Response 6: Thank you for pointing out the serious mistake. We have changed the scale bar unit in kilometers (km).
In the end, thanks a lot for your important suggestion.
With best regards
The authors

Reviewer 2 Report
In this study the authors used satellite Landsat 8 data, and ground-based air quality and meteorological data to assess spatial PM2.5 concentrations in the region of Xi’an, Shaanxi, China. Based on these integrated data obtained for the period 2017-2021. standard ML algorithms Multi-Layer BackPropagation Neural Network (MLBPN), Support Vector Regression (SVR), and RandomForest (RF) were applied for PM2.5 assessment. According to the standard evaluation indices MLBPN showed the best PM2.5 estimation result with the additional Kriging interpolation method for error correction. In addition, the representativeness of the monitoring stations in the region was further evaluated based on the annual average comparison between measured and assessed results. The methodology can be used for new monitoring sites selections improvement to obtained more reliable and regional representativeness of ground stations network.
Generally, the paper is interesting although there are many published papers related to this specific problem of air pollution assessment. Methodology part can be improved and some aspects can be omitted since the standard already well-known ML algorithms were used. The authors used public available data and standard ML methodology. Having in mind the characteristics of the region studied and some aspects of error analysis, I found the obtained results useful.
Please find below several suggestions that might be consider for paper improvement:
- In the methodology part short description of B1, B3, B7 bands used should be added. Please explain physical background why these band are relevant for ground based PM concentrations but not others?
- Air quality data is not precise term since only PM2.5 data were used; it would be better to use only PM2.5 data. Other air quality data from measurements stations were not used in this study.
- From fig.1 it can been seen the distribution of 13 measurement stations: it seems grouped (localized) – please explain their representativeness for that pretty large area showed in maps
- Meteorological data were obtained from CMA-NOAA. It is stated that only one station is in this region (Fig 1.). It not clear whether the authors used measured data from this one station only, or other modeling data? Please comment on this and representatives of meteorological data. Can authors provide some kind of error assessment of these data since as input parameters in ML algorithms systematic errors is introduced.
- What are the criteria for variables (input parameters) used in ML models? Are the same input parameters used for each model? If so, why? There is feature selection methodology widely used so that most important parameters can be used in modeling to achieve best performances. Usually, better results can be obtained if some of the parameters are not selected as input variables when applying specific ML algorithm. Please comment on this issue.
- Regarding the validation procedure: there is no information what is the total number of data used? Since the satellite data have 16 days time resolution, please explain how the data fusion was performed. What is the number of training and validation data?
- Lines 330-334: This information is already presented in Table 2, no need to mention it twice, it’s some kind of redundancy.
- Figure 8 and 9. Please add units
- Due to the consistency, please use terms estimated and measured (not row data and output prediction – like in legend in Fig 9)
Author Response
Thanks a lot for taking the time to review our article. Your comments and suggestions are very valuable and helpful for the improvements of this manuscript. According to the comments/suggestions of all of the reviewers, the manuscript has been substantially refined from both technical and linguistic points of view, marked using the “Track Changes” function in the attached word-file. Moreover, the final version after the revision (without the tracked changes, and marked by the ‘highlight color’) can be found in the attached PDF-file
By the way, please see below our point-to-point responses to your comments.
Point 1: In the methodology part short description of B1, B3, B7 bands used should be added. Please explain physical background why these band are relevant for ground based PM concentrations but not others?
Response 1: Thank you for pointing them out. According to some literatures published so far, the Band 1, 3, 7 besides the calculated NDVI based on the Bands 4 and 5 from Landsat 8 OLI, are strongly relevant to the PM2.5 concentration. Hence, the Band 1, 3, 7 and NDVI of Landsat 8 have been employed as the primary source data in our research. In the revised manuscript, we have appended the relevant descriptions (please ref. the sentence of “Based on some literatures published so far [22,25], B1 (0.43-0.45 µm), B3 (0.53-0.59 µm), B7 (2.11-2.29 µm) and the calculated NDVI based on the B4 (0.64-0.67 µm) and B5 (0.85-0.88 µm) of Landsat 8 are strongly relevant to the PM2.5 concentrations. Therefore, B1, B3, B7 and the NDVI of Landsat 8 have been selected as the primary input parameters for the PM2.5 concentration estimations.” in the end of section “2.2.1 Satellite Remote Sensing Data”. Hopefully, it could become clearer.
- Zhang, B.; Zhang, M.; Kang, J.; Hong, D.; Xu, J.; Zhu, X. Estimation of PMx Concentrations from Landsat 8 OLI Images Based on a Multilayer Perceptron Neural Network. Remote Sens. 2019, 11, 646. https://doi.org/10.3390/rs11060646.
- Li, R.; Sun, L. An Improved DDV Method to Retrieve Aerosol Optical Depth for Landsat 8 OLI Image. Remote Sens. Inf. 2017, 32, 120–125. https://doi.org/10.3969/j.issn.1000-3177.2017.01.021.
Point 2: Air quality data is not precise term since only PM2.5 data were used; it would be better to use only PM2.5 data. Other air quality data from measurements stations were not used in this study.
Response 2: Thank you for your valuable suggestion. We have changed Air quality data to PM2.5 data.
Point 3: From fig.1 it can been seen the distribution of 13 measurement stations: it seems grouped (localized) – please explain their representativeness for that pretty large area showed in maps.
Response 3: As you pointed out, the 13 measurement stations are mainly located in the built-up area, which can affect the estimation results of the PM2.5 concentrations in the whole Xi’an area. According to our analysis in this manuscript, the 13 monitoring stations themselves cannot represent the whole Xi'an region. That’s why we use satellite remote sensing images for the PM2.5 estimation in the Xi’an area -- by combining different bands of Landsat 8 and meteorological factors, we establish a mapping relationship with PM2.5 and obtain the overall PM2.5 distribution. The representation of 13 stations is also analyzed in the section “3.3 Representativeness Analysis”.
Point 4: Meteorological data were obtained from CMA-NOAA. It is stated that only one station is in this region (Fig 1.). It not clear whether the authors used measured data from this one station only, or other modeling data? Please comment on this and representatives of meteorological data. Can authors provide some kind of error assessment of these data since as input parameters in ML algorithms systematic errors is introduced.
Response 4: Thank you for pointing them out. Due to the different number of monitoring stations and geographic locations, it is complex to directly obtain the meteorological information around the CNEMC stations. Instead of using only the values of one station, we obtain the meteorological data of CNEMC station by interpolating the data of the surrounding CMA-NOAA stations. According to our study, we can still get satisfied estimated results without meteorological factors, so we judge that the satellite bands are the main influencing factors. Therefore, the error caused by meteorological interpolation will not affect the estimated results too much.
Point 5: What are the criteria for variables (input parameters) used in ML models? Are the same input parameters used for each model? If so, why? There is feature selection methodology widely used so that most important parameters can be used in modeling to achieve best performances. Usually, better results can be obtained if some of the parameters are not selected as input variables when applying specific ML algorithm. Please comment on this issue.
Response 5: Thank you for pointing them out. The same input parameters are used for each model, because we have tried to explore the optimal combination of the influencing factors, but the current input parameters are still the optimal combination. The steps are as follows:
According to the significance of the correlation to the PM2.5, as well as the different data sources and characteristics, the 17 alternative input influencing factors were categorized into the following three groups, viz.:
- a) the reflectance of bands 1, 3, 7 and the calculated NDVI based on the bands 4 and 5 from Landsat 8 OLI, which have a strong correlation with the PM5 concentrations, abbreviated as strong correlation band (SCB);
- b) the reflectance of the other bands from Landsat 8 OLI, abbreviated as “OB”;
- c) the meteorological parameters of PRS (Pressure), RH (Relative Humidity), TEM (Temperature), WD (Wind Direction), and WS (Wind Speed)., abbreviated as “MP.”
As the parameters in the group of “SCB” are strongly correlated to the PM2.5 concentrations and thereby treated as the “basis” for the PM2.5 estimation, which will be employed throughout the whole process to explore the optimal combination of the alternative input influencing factors. In order to find the optimal combination of the alternative input parameters for the PM2.5 estimation model, the parameters in these three different groups were stepwise input to the PM2.5 estimation model for training, learning, and validation.
The training results generated by different combinations of the alternative input parameters can be compared according to ME, RMSE, and R2. Thus, the optimal combination of the input parameters with respect to the various influence factors can be achieved.
Point 6: Regarding the validation procedure: there is no information what is the total number of data used? Since the satellite data have 16 days time resolution, please explain how the data fusion was performed. What is the number of training and validation data?
Response 6: A total of 974 samples were obtained, with the training set accounting for 80% and the validation set accounting for 20%.
In the process of geospatial analysis, it is essential to ensure that all of the relevant elements are at the same or very approximate temporal stamps. The imaging time property of the Landsat 8 data is regarded as the temporal ‘reference’ for the conducted research. Due to the different time periods of the CNEMC and CMA-NOAA measurements, the time buffer intervals of ±0.5 h and ±1.5 h are settled for data from CNEMC and CMA-NOAA, respectively, using linear interpolation to obtain the required parameter datasets. In this way, the data from Landsat 8, CNEMC and CMA-NOAA can be harmonized together.
Point 7: Lines 330-334: This information is already presented in Table 2, no need to mention it twice, it’s some kind of redundancy.
Response 7: Thank you for your insightful suggestion. We have deleted the sentences.
Point 8: Figure 8 and 9. Please add units
Response 8: Thank you for pointing out the serious mistake. We have added units in these figures.
Point 9: Due to the consistency, please use terms estimated and measured (not row data and output prediction – like in legend in Fig 9)
Response 9: Thank you for your valuable suggestion. We have changed the legend.
In the end, thanks a lot for your important suggestion.
With best regards
The authors

Round 2
Reviewer 1 Report
I have no more issues.
Author Response
Thanks a lot for taking the time to review our article and your encouraging comment. Your comments and suggestions are very valuable and helpful for the improvements of this manuscript. According to the comments and suggestions of all of the reviewers, the manuscript has been substantially refined from both technical and linguistic points of view.
In the end, thanks a lot for your important suggestion.
With best regards
The authors
Reviewer 2 Report
I have read the revised manuscript which is moderately improved according to most of the suggestions. The authors added several explanations and reasonably addressed previous issues.
I’m not against publishing after considering several technical details:
- Figure 7 is not visible (might be due to the pdf conversion process?)
- Several typing errors should be corrected i.e. lines 293, 295,…
Author Response
Thanks a lot for taking the time to review our article. Your comments and suggestions are very valuable and helpful for the improvements of this manuscript. According to the comments/suggestions of all of the reviewers, the manuscript has been substantially refined from both technical and linguistic points of view, marked using the “Track Changes” function in the attached word-file. Moreover, the final version after the revision (without the tracked changes, and marked by the ‘highlight color’) can be found in the attached PDF-file
By the way, please see below our point-to-point responses to your comments.
Point 1: Figure 7 is not visible (might be due to the pdf conversion process?)
Response 1: Thank you for pointing them out. We have revised the manuscript and now Figure 7 is visible in both of the pdf and word file.
Point 2: Several typing errors should be corrected i.e. lines 293, 295,…
Response 2: Thank you for pointing out the serious mistake. We have checked the manuscript and corrected the typing errors. E.g.:
“esti-mated” to “estimated” in Line 285
“superim-pose” to “superimpose” in Line 286
In the end, thanks a lot for your important suggestion.
With best regards
The authors
